# Curcumin May Prevent Basement Membrane Disassembly by Matrix Metalloproteinases and Progression of the Bladder Cancer

**DOI:** 10.3390/nu14010032

**Published:** 2021-12-23

**Authors:** Paweł Wroński, Stanisław Wroński, Marcin Kurant, Bartosz Malinowski, Michał Wiciński

**Affiliations:** 1Department of Pharmacology and Therapeutics, Faculty of Medicine, Collegium Medicum in Bydgoszcz, Nicolaus Copernicus University, M. Curie 9, 85-090 Bydgoszcz, Poland; wronaone@gmail.com (P.W.); bartosz.malin@gmail.com (B.M.); 2Department of Oncological Urology, The Franciszek Lukaszczyk Oncology Center, Romanowskiej 2, 85-796 Bydgoszcz, Poland; 3Department of Urology, Jan Biziel Memorial University Hospital, Ujejskiego 75, 85-168 Bydgoszcz, Poland; s.wronski@icloud.com; 4Department of Urology, District Hospital, 10 Lesna Street, 89-600 Chojnice, Poland; marcinkurant@o2.pl

**Keywords:** bladder cancer, basement membrane proteins, phytochemicals, curcumin, matrix metalloproteinases

## Abstract

Authors present a review of crucial mechanisms contributing to the invasion of the basement membrane (BM) of the urothelium by cancer cells and to the progression of bladder cancer (BC). The breeching of the urothelial BM, facilitated by an aberrant activation of matrix metalloproteinases (MMP) is particularly perilous. Inhibition of activation of these proteinases constitutes a logic opportunity to restrain progression. Because of limited efficacy of current therapeutic methods, the search for the development of alternative approaches constitutes “the hot spot” of modern oncology. Recent studies revealed significant anticancer potential of natural phytochemicals. Especially, curcumin has emerged as a one of the most promising phytochemicals and showed its efficacy in several human malignancies. Therefore, this article addresses experimental and clinical data indicating multi-directional inhibitory effect of curcumin on the growth of bladder cancer. We particularly concentrate on the mechanisms, by which curcumin inhibits the MMP’s activities, thereby securing BM integrity and alleviating the eventual cancer invasion into the bladder muscles. Authors review the recently accumulating data, that curcumin constitutes a potent factor contributing to the more effective treatment of the bladder cancer.

## 1. Introduction

Bladder cancer (BC) is the fourth most common malignancy among men and the eighth most common in women, and accounts for 5–10% of all cancer cases [1,2]. Because of the high recurrence rate, total costs of the treatment are the highest from all malignancies [2,3,4,5]. The most common type of bladder cancer is urothelial carcinoma arising from stratified epithelium-urothelium. Layers of cells are arranged into strata perched on the basement membrane (BM), then on lax connective tissue, followed by the muscular part of the bladder wall [6]. The BM and underlying lamina propria can be regarded as a strategic field of confrontation with invading cancer cells, and both definitely influence the course of disease. Further, the components of both membranes are targeted during bladder cancer treatment. At the time of initial diagnosis, 70–75% of cases present as superficial non-muscle-invasive tumors (NMIBC). The rest of cases present as lamina propria breaching, muscle-infiltrating lesions [1,2]. Five-year survival for NMIBC exceeds 80%. Prognosis significantly decreases with depth of infiltration. For MIBC, 5-year survival is up to 17–57% and 20% when the tumor crosses the bladder wall [7,8]. This imposes the development of new efficient methods to treat bladder cancer in an early urothelium-confined phase, and to prevent BM decomposition with further transition into deadly muscle-invasive and metastatic stages. The following paper is focused on the processes and factors modifying the basement membrane and subepithelial connective tissue (lamina propria), and on possible cost-effective methods of bladder cancer therapy. Matrix metalloproteinases, a class of zinc-dependent enzymes, form decisive biologically active complex modifying above structures. By that rationale, the authors analyzed possible modification of metalloproteinases’ activity to significantly reduce the risk of bladder cancer progression. The authors focused on a possible practical application of a biologically active natural nutrient—curcumin—as a new frontier in the effort to reduce bladder cancer mortality. To fully elucidate and to make this review understandable, the structure and function of the basement membrane, matrix metalloproteinases’ characteristics, and a description of curcumin biological properties will be presented at following sections.

## 2. Basement Membrane Structure, Function and Pathological Modifications

There is a sometimes-neglected structure protecting the conversion of histopathological cancer into cancer disease. This line of defense is the basement membrane. Factors disrupting this protecting last frontier favor the progression of cancer, but may also be a target for novel anticancer therapies. A specific group of enzymes degrading basement membrane proteins is of special interest. Bladder cancer is an illustrative example.

As other epithelia, the urothelium is placed on the basement membrane (BM). The basement membrane is a specific sheet-like structure separating different compartments: epithelial cells from underlying connective tissue, in the case of the bladder wall from lamina propria (Figure 1) [9,10]. 

It is hard to define the BM structure as uniform for all tissues, because it may have diverse roles in different organs, and because of the variability of distribution of different isoforms of basic proteins [11]. Up to 200 proteins are described in the BM in different localizations. The majority have unknown function, and their reciprocal influence and interdependence are not elucidated [12]. However, the basic BM pattern is common in all cases. The BM is composed of two merged, but distinct, networks of fibrous proteins: laminins; and a basement-membrane-specific collagen IV, which in consort with extracellular matrix binding proteins—nidogens and heparan sulfate proteoglycans (HSPGs: collagen XVIII, perlecan, and agrin)—forms the core structural scaffold decisive for BM integrity [13]. Several other components are located in the BM from different tissues, depending on their specific local role. Such tissue-specific membranes contain dedicated subtypes of laminin, collagen, and glycoproteins as a real fingerprint for a given type of epithelia (Figure 1) [10]. 

Both collagen IV and laminin frameworks form stable support and a barrier for cells. However, the size and charge of defined pores allow for selective permeability and filtration of vital molecules [13,14,15]. Structural components of BM connect with urothelial cells through corresponding cell membrane receptors: integrins, dystroglycans, and discoidin domain receptors (DDRs). Through reciprocal interactions with the epithelium, the basement membrane contributes to the cellular differentiation and polarity, and to the establishment of tissue homeostasis and coherent architecture [14,16]. The status of the BM is not permanently fixed, but subject to changes even under normal conditions. Modifications and remodeling of previously spatially-restricted structures are crucial for early malignant development, and further cancer progression. 

As a result of the above, some novel anticancer strategies should target enzymes degrading BM components. Zinc-dependent matrix metalloproteinases (MMPs), a family of proteases degrading the normal extracellular matrix, play a key role in these processes. MMP are also found in all primary and recurrent tumors [9]. Two proteases, MMP-2 and MMP-9, are regarded as unique for specifically targeting basement membrane proteins. This activity goes bidirectional: mechanistic, by degrading the physical barrier between tissues and the undermining fulcrum for proper epithelial architecture; and biochemical, by proteolysis, generating fragments with biological activity, or unchaining tethered growth factors, its inhibitors, and other regulatory macromolecules [10]. Aberrant regulation of MMP-2 and -9 has been found during malignant transformation, and further progression of invasive types of bladder cancer and several other neoplasms [17]. 

For example, MMP-2 and -9 cleave laminin and collagen IV, respectively, releasing biologically active fragments influencing essential cancer-related processes as angiogenesis and epithelial-to-mesenchymal transition (Figure 1) [18,19]. MMPs inhibitors tested so far did not find a clinical application due to the lack of specificity only for tumor tissues [14]. However, a number of substances contained in natural nutrients show distinct anti-tumor and anti-MMPs activity, and may find clinical applications. 

Laminin is the main and most abundant glycoprotein of BM. It determines growth of epithelial cells, as well as molding extracellular microenvironment, forming specific niches essential for cell contact-dependent survival [20]. Normal cells which lose contact with laminin and its location in the niche inevitably undergo apoptosis. However, when the expanding cancer cells supersede mesenchymal cells, normal laminin production decreases. This leads to BM disintegration, and facilitates cells translocation into underlying tissue layers (e.g., urinary bladder wall) [21]. Laminins are also components of hemidesmosomes’ stable adhesions anchoring cells to the extracellular matrix [14]. These adhesions allow resilience for shear forces between different tissue layers when the urinary bladder changes volume [12,14]. 

Collagen IV is another cross-linked fibrous protein characteristic to the BM. The collagen IV network locates mainly at the stromal face of the BM, and binds with cells through collagen-binding integrins and cell membrane discoidin domain receptor 1 (DDR1) [12]. This mesh constricts growing epithelia, and thus, shapes tissues [12]. Similar to laminin, besides mechanical anchoring, neighboring structures provide a pathway for signal transduction for cell proliferation and morphology [12]. 

Both cross-linked networks are mutually influential, and associate with several ECM proteins. Two proteins, nidogen (glycoprotein) and perlecan (heparan sulfate proteoglycan), bridge these networks together. Also, both networks anchor neighboring cells through interaction with receptors and sulfated glycolipids on the cell surface [12,13]. The complexity of interplay of all known (and unknown) factors illustrate how BM and separated tissues are interlocked [12,13] (Figure 1). 

Nidogens (type 1 and 2), also known as enactins, are ubiquitous monomeric BM glycoproteins synthetized by mesenchymal cells. Their high affinity to laminin and collagen IV promotes bridging of these two basal self-assembled networks into a uniform scaffold [13]. Nidogen also binds with adhesion receptors, anchors cells to the matrix, and promotes cell polarization [12].

Heparan sulfate proteoglycans (HSPG), perlecan, agrin, and collagen XVIII, are also key, specific molecules of the extracellular matrix [10]. They bind together ECM proteins, thus, defining its structure; have prominent hydrophilic capacity, thus, defining its volume; and bind and tether growth factors, thus, defining differentiation. In effect, proteolysis of HPSGs frees molecules essential for normal tissue development, but also crucial for malignant formation [15,22]. Other proteins related to the BM—interstitial collagens (V and VII), stromal collagens (I and III), elastins, and fibronectins—structurally bind cells, regulate cell–matrix interactions and cell morphology, and tether other ECM proteins [13]. Urothelial cell membrane receptors—integrins, DDRs, and dystroglycans—associate with interwoven laminins and collagen IV networks, and anchor cells to the BM. In the tumor microenvironment, they regulate and facilitate the interaction of cancer cells with the collagen network, and contribute to cancer progression. Their abnormal function occurs in several human benign and malign pathologies (Figure 1) [23,24,25].

Critical for BM proper maintenance and remodeling, but also for BM pathology, is a diverse family of zinc-dependent proteinases called matrix metalloproteinases (MMPs). Matrix metalloproteinases are widely present in the BM in soluble, and protein- and cell-membrane-bound forms [12,13,23]. Multiple types of molecules and cytokines are docked in ECM structures; thus, their proteolytic degradation by MMP activates and releases them into the microenvironment (Figure 1) [23].

## 3. Alterations of BM Structure during Malignant Processes

At the beginning of bladder tumor formation, the intact basement membrane separates and confines neoplastic cells from underlying connective tissue [14]. To cross, BM invasive cells must degrade this barrier [13]. At advanced stages, deep invasion is related to specific and substantial remodeling of the ECM [13]. 

The most characteristic feature of the early stages of malignant transformation of the urothelium is the distortion of normal architecture, and the loss of cellular polarity, correlating with tumor aggressiveness [9,14]. This is due to alterations of receptors and hemidesmosomes for BM proteins located on neoplastic cell membranes [9]. Altered integrins lose their role as a determinants and keepers of tissue integrity, and promote further progression. These receptors are redistributed to cell membrane protrusions (invadopodia) to stimulate migration, adhesion, and survival into invaded tissues (Figure 1) [26]. 

Actin-rich invadopodia of the malignant cells determine their migration and traversal through BM pores by expanding regular micropores, and by degrading their elements, mainly MMP-2 and -9. [12,14,27]. Protrusions can extend deep into surrounding connective tissue, forming contacts with mesenchymal cells, and adhesions fixing to the underlying stroma [14]. This process is similar to implantation of embryonic trophoblasts [12]. BM degradation goes through proteolysis by MMP, which, in turn, have several sources: aberrant synthesis and secretion by cancer cells themselves, from cancer-associated macrophages attracted by cytokines secreted by cancer cells (IL-6, IL-2), and from inflammatory clusters in tissues adjacent to the tumor [28]. Degree of proteolytic degradation of BM by MMPs is usually concordant with tumor dedifferentiation [29]. Malignant transformed epithelial cells, which can traverse BM, also undergo morphological conversion, gaining some mesenchymal traits, such as: motility, loss of intercellular adhesions, and polarity. This phenomenon is called epithelial-to-mesenchymal (EMT) transition, and allows for cancer cells to transfer to distant organs to form metastases. MMP-9 and -2 expose specific segments of laminin and collagens, called cryptic fragments, with proangiogenic features, and propagate epithelial-to-mesenchymal transition [12]. At some point, EMT recapitulate processes met during embryonal organogenesis [30,31]. 

Bladder tumor cells not only degrade and decompose invaded BM, but also “creatively” rebuild it. Excessive accumulation of ECM structural proteins causes extensive fibrosis of infiltrated tissues, called desmoplasia. Desmoplasia is found in high grade bladder cancers clinically presented as a solid fibrous bulk of tumor [13]. Next, highly metastatic tumors secrete laminin and collagen IV by themselves, and overexpression of both proteins is related to hyperproliferation and tumor angiogenesis [12]. During further invasion, cells attain a muscle-invasive characteristic, and gain the capacity to metastasize [10,14]. Tumors synthesize their own set of enzymes, degrading collagenous and non-collagenous substrates to pass the basement membrane, collagenases, and heparinases, respectively [13]. The expression of tumor-derived collagenases correlates with tumor aggressiveness, and is augmented according to its metastatic potential. Cells with high metastatic potential degrade heparan sulfate at higher rate than malignant cells with low metastatic potential [13].

## 4. Basement Membrane and Specific Role of MMP-2 and -9 in Urinary Bladder Cancer Progression

All processes involved with normal organ development and remodeling, and with all pathologies are entirely dependent on BM reorganization. BM degradation is recognized as a hallmark of cancer [12]. Cleavage of the BM is indispensable for tumor invasion and progression for several reasons: first, to clear the way for breaching and crossing the mechanical barrier; secondly, to alter epithelial cells’ polarity, enabling EMT; thirdly, for releasing active fragments of BM molecules such as previously tethered growth factors, or for exposing cryptic fragments of ECM proteins with signaling functions; fourthly, for modulation of the signaling pathways [32]. This is achieved by means of matrix metalloproteinases required for the degradation of BM components [15,23,33]. A large family of zinc-dependent proteases comprises up to 23 members with different tissue and substrate specificity. In basement membranes and in the extracellular matrix, MMPs exist as free or protein- or membrane-bound forms [23]. Two MMPs—MMP-2 and -9—are BM-specific because of their localization and activity. Both enzymes degrade BM elements, but to conduct its activities, MMPs must be transformed into an active form [23]. MMPs are synthesized as latent propeptides which must be activated by cleavage of an inhibitory fragment. This runs both intracellularly by furins, and extracellularly in ECM milieu. Cleavage is performed by several different enzymes, or autoproteolysis by other MMPs. MMPs are generated by cancer cells themselves, by inflammatory cells occupying surrounding environment, or cleaved from tethering ECM molecules [14,19]. A special type of cell-membrane-bound metalloproteinase anchored on the cell surface (MT-MMPs) activates soluble MMPs in the immediate proximity to the given cell. Additionally, signaling molecules secreted by inflammatory cells invading the tumor microenvironment can induce and upregulate MMPs at the gene and protein level [34]. This process runs in tumor cells during malignant infiltration of the BM [23,35]. Activated MMP also undergo regulation in the tumor microenvironment, primarily through binding by specific inhibitors of metalloproteinases (TIMPS). Yet, to make things more complicated, TIMPS can activate MMP propeptides, and propagate cell proliferation, similar to growth factors [35]. 

During invasion, newly formed invadopodia force through BM pores. Integrins are relocated, form clusters on these protrusions, and bind MMPs for targeted focal proteolysis of the surrounding matrix in the most strategic locations [36]. During this stage, membrane-type 2 MMP (MT2-MMP) not only facilitates BM trafficking, but also additionally releases active soluble fragments of collagen XVIII (endostatin) [37]. Endostatin, a highly active peptide, exerts a plethora of different activities: it stimulates migration and proliferation of endothelial cells, and modulates angiogenesis and morphogenesis [38,39]. Endostatin is an inhibitor of pathologic tumor angio- and lymphangiogenesis which downregulates tumor-derived vascular endothelial growth factors (VEGFs) and its receptors, and inhibits nodal and distant metastasis [33]. Further tumor invasion depends mainly on MMPs from the surrounding microenvironment where enzymes are bound to proteins, or secreted by normal cells [40,41]. Activated MMP-9 by the degrading collagen IV generate derived molecules, regulating angiogenesis [18]. Next, laminin proteolysis by MMP-2 unpins fragments, stimulating epithelial-to-mesenchymal transition, which facilitates and propagates cell motility and tissue infiltration [19]. Proteolysis of perlecan (another BM key protein) generates more pro-invasive and adhesion-stimulating molecules localized in advanced tumors [42,43,44,45]. Endostatin downregulates MMPs, and inhibits above proteolytic activities (Figure 1) [33]. Patients with Down syndrome are characterize by a low incidence of solid tumors, due to elevated levels of endostatin related to the extra copy of the 21st chromosome carrying gene for collagen XVIII [46].

## 5. Biological Properties of Curcumin in Cancer Processes

For ages, herb- and plant-derived medicines have been a staple in therapy, even without exact knowledge of contained active components or the mechanism of action. Several extracted pure substances, their derivatives, or synthetic equivalents are widely used even today. Contemporary molecular biology and analyses of cell signaling pathways reveal mechanisms of action, and potentially new pharmacological uses of known old nutrients present in common food. Natural ingredients from food (nutrients, nutraceuticals, phytochemicals) or their derivatives can replace or synergistically reinforce the action of current medicines when combined with standard therapeutic regimens [47]. 

New findings provide a very promising perspective of selected compounds as substances with high therapeutic potential for human malignancies. The potential is promising for 2000 plants, each containing numerous molecules, which are under laboratory and clinical evaluation now [48]. Despite their demonstrated anti-cancer efficacy, the precise molecular mechanism of activity is not clearly established [49]. The activity of phytochemicals tested in vitro on experimental cell lines differs from effects observed on clinical settings for the multiplicity of additional interactions and interdependencies between different cells and tissues [50]. Last but not least, relatively inexpensive and easily available natural phytochemicals and derivatives may be difficult to compete with expensive, extolled drugs. 

An example of a nutrient exerting potent biological activity is curcumin, currently the most intensively tested phytochemical. Widely used as a food additive, it shows a lack of any toxicity. [51,52,53,54]. Its curative properties, as used in traditional Asian medicine for the treatment of nonhealing wounds and gastrointestinal diseases, present safety even in high doses [55,56]. Recently, curcumin gained attention due its antioxidant, antiatherosclerotic, anticancer (antiproliferation, anti-invasive, and antimetastatic) activity [34]. 

Curcumin is an active compound from the turmeric rhizome Curcuma longa. Raw or dried-pulverized turmeric is commonly used as a spice in Asia [55,57]. Up to 235 bioactive compounds have been extracted from turmeric. The most abundant are curcuminoids: curcumin, demethoxycurcumin, and bisdemetoxycurcumin, in a proportion 7:2:1, respectively [58]. Nowadays, curcumin is used in food processing as a spice and as a natural pigment. Its chemical structure was first described by Polish chemists in 1910 [59]. Chemically, curcumin is a polyphenol-diferuloylmethane: (1,7-bis(4-hydroxy-3-methoxyphenyl)-1,6-heptane-3,5-dione) [52,55]. It has two constitutional isomers: enol and b-diketone tautomeric forms. The former exists predominantly in solutions, the latter is important for free radicals’ scavenging ability [55,60]. 

The degradation of the basement membrane—a prerequisite for tumor invasion—is mediated mainly by MMP-2 and -9. Abundant data present that the anticancer and antimetastatic effect of curcumin is linked to inhibition of MMP activity. This potential of curcuminoids is dose- and time-dependent, as demonstrated on in vitro cultured cells [55,61]. In addition to downregulation of MMPs, curcumin upregulates expression of tissue inhibitors of MMP, especially TIMP-2. Further, curcumin counteracts metastasis formation by the restriction of cancer cell adhesion molecules, allowing for binding to ECM [62]. Curcumin-related decreased expression of MMP-2 and -9 also inhibits angiogenesis [63]. The mechanisms of such an impressive range of activities have not been fully elucidated yet. Diverse anticancer and antioxidant properties seem to have common crucial elements linking these activities. Curcumin inhibits cell signaling pathways, and hampers expression of several cancer-related genes (for instance, COX-2, TNF, cyclin D1). Several papers present that its main impact point focuses on nuclear factor kappa B (NF-κB) and STAT pathways regulating the above genes (Figure 2) [55]. NF-κB is a ubiquitous inducible transcription factor present in all animal cells, and is the pivotal element of the pathway transmitting extracellular signals into the nucleus to stimulate the expression of numerous genes. Thus, curcumin appears to be a master regulator of almost all cellular processes involved in cell proliferation, survival, and response to external factors. It binds to promotors of targeted genes, and activates their transcription [64]. Substances which can control and modulate NF-κB, in turn, can control and modulate the function of genes and the cell’s fate. Curcumin has properties to suppress the activation of NF-κB (Figure 2). Binding any ligand to the cell surface receptor induces a specific kinase, IKK, which phosphorylates, and thus, inactivates an inhibitor of NF-κB-IκB. The promoters of MMP genes have binding sites for NF-κB. Then, NF-κB dimerizes and translocates to the nucleus to promote transcription of any given gene for cell membrane-bound metaloproteinase MT1-MMP. MT1- MMP proteolyse pro-MMP-2 into active MMP-2. Curcumin interrupts this pathway by blocking IκB kinase (IKK), and thwarts MMP-2 activation, and subsequent extracellular matrix degradation and invasion [65]. Concurrently, other NF-κB dependent genes and their products (MMP2,9, cyclin D1, CoX-2, Ras, iNOS, bcl-2, bclxl) are downregulated and suppressed, so cell proliferation and migration are inhibited. A similar protecting mechanism of curcumin has been described for restricting NF-κB benzopyrene (a cigarette smoke compound)-related activation of genes expression in selected lung carcinoma cell lines [66]. Additionally, curcumin through bcl-2/bclxl-related inhibition of NF-κB pathways activates caspase-mediated cell death [66,67]. An alternative pathway of curcumin suppression on MMP-2 runs through the Akt serine/threo-nine protein kinase axis, as proved by curcumin’s anti-lymphangiogenic effect on an experimental line of lymphatic endothelial cells (Figure 2) [67]. 

A probable mechanism of MMP inhibition relies on zinc ions locking through the metal-binding moiety of curcumin [55,56]. Another inhibitory mechanism of curcumin on MMP-2 and -9, and on proliferation signaling pathways is in the decreasing expression of extracellular signal-regulated protein kinase (p-ERK1/2), associated with cell proliferation and survival. A transcription factor regulator p-ERK1/2 is involved in signal transduction from the cell membrane to the nucleus [55,68]. A mammalian target of rapamycin (mTOR) pathways is a next target for curcumin inhibitory activity, and through the phosphatidylinositol 3-kinase/Akt/IkBk kinase axis, MMP-2 and -9 expression are also significantly suppressed [55]. Further, curcumin also selectively targets MMP2 by restricting expression of mRNA for MMP2 [69]. Curcumin-induced apoptosis was experimentally presented in prostate cancer cell lines as propagated through TNF-a-related apoptosis-inducing ligand (TRAIL). 

Because NF-κB is a component of pro-inflammatory signaling pathway, and is involved in the production and release of inflammatory cytokines such as TNF-alfa, which, in turn, activate MMP, this mode of activity can also potentiate the antitumor properties of curcumin [70]. Tumor-infiltrating lymphocytes express upregulation of MMP-9, which, in turn, promotes neoplastic proliferation and invasion. Curcumin inhibits this trait by suppressing the cancer propagatory influence of cytokines excreted from the inflammatory transformed peritumoral extracellular matrix [71]. Further, decrease of MMP-2 expression, and an increase of TIMP-1 intensified antiproliferative and anti-invasive effects on a breast cancer cell line [55,71]. Another mechanism of indirect downregulation of MMPs expression relies on the ability of curcumin to suppress ECM inducer (EMMPRIN, CD147), crucial for cancer cell adhesion, tumor invasion, and metastasis. This cell membrane glycoprotein is abundant in cancer cells, and stimulates production, secretion, and activation of MMPs by cells (monocytes and macrophages) present in tumors surrounding the EC stroma. CD147 also binds MMP at the cell surface, and helps the cell to directly degrade surrounding pericellular ECM and further invasion [72]. Curcumin inhibition of MMP-2 and -9 activity also restores blood–brain barrier integrity, as reported for cerebral ischemic injury [34]. Another positive involvement on vascular pathology has been presented. Curcumin reduces activity of MMP-2 and -9 during plaque formation, and suppresses NF-κB and MMP-9 expression in vascular smooth muscle cells, contributing to atherosclerosis formation (Figure 2) [55]. 

Unfortunately, important issues related to the use of curcuminoids still remain unresolved. Clinical application of curcumin is hampered by its somewhat unfavorable biological properties. Bioavailability of curcumin is low because of its limited water solubility at acidic and neutral pH, resulting in poor intestinal absorption, and also its rapid degradation by glucuronidation in the intestinal wall and in the liver [53,73]. To overcome barriers related to the limited bioavailability of curcumin, a number of modifications to the administration or its chemical structure have been tested. Research has been aimed at improving the absorption of curcumin, on slowing its metabolism, and on linking it with other substances to achieve synergy; thus, enhancing their therapeutic activities. Co-administration with piperine enhances intestinal absorption, and diminishes biodegradation (glucuronidation) of curcumin, and thus, increases curcumin bioavailability up to 20× [51]. Curcumin-piperine formulation as a nanoparticle is a useful solution, and has been extensively tested. [74,75]. The combination with phosphatidyl-choline increases oral absorption of curcumin in humans 5-fold [55]. Another formulation combines both molecules into liposomes as one stable water-soluble delivery system [51]. Experimental delivery systems couple curcumin with chemotherapeutics [76,77]. Liposomal formulation with polyethylene glycol, and complexes with phospholipids or dextrin have been tested for intravenous administration [51]. The addition of hydrophilic groups considerably improves the solubility of curcumin [75,76,77,78]. To make curcumin more relevant for bladder cancer treatment, intravenous infusion or intravesical instillations have been tested for the prevention of recurrence after tumor resection or BCG-therapy [77,78]. Other solutions enhancing curcumin absorption consider metal complexes, magnetic microspheres, or solid-lipid nanoparticles, which can be delivered straight to the targeted organs [74,75,76]. Published clinical trials present that complexing cyclodextrin with curcumin significantly improves intestinal absorption of curcuminoids [75,78]. Such commercial preparations are now available (brand name: Meriva, Curarti) [79,80].

Another field of clinical application is the synergistic activity of curcumin with existing anticancer agents. Co-administration of curcumin and paclitaxel significantly reduced expression of metalloproteinase-2, and decreased paclitaxel side effects in a PC3 xenografted prostate cancer model. In effect, such a formulation is proposed for hormone-refractory prostate cancer (HRPC) [55]. The synergistic action of gemcitabine plus curcumin has been presented in in vivo and in vitro models of pancreatic cancer [81]. In pancreatic cancer cell lines, inhibition of proliferation and apoptosis were substantially increased. Mice models of pancreatic cancer presented a significant reduction of tumor volume, suppression of NF-κB regulated genes (for cyclin D1, Bcl-2, BclxL, COX-2, matrix metalloproteinases, VEGF), and decreased microvessel density (Figure 2) [81]. 

Structural modifications of the curcumin molecule may facilitate its bioavailability, and increase treatment potential. New chemical analogues of curcumin form a very promising group of anticancer drugs. An analog FLLL32 blocks binding of transcription factors to DNA; induces its degradation; decreases VEGF, MMP-2, survivin expression; and promotes apoptosis in an experimental human osteosarcoma cell line [82]. Another analog is formulated by altering aromatic moiety to increase water solubility, metal binding, and MMP inhibition [50]. Contemporarily, a few dozen curcumin derivatives and dibenzoyl analogs have been tested for possible antioxidant, antiproliferative, and anti-inflammatory efficacy [50]. Hydrazinocurcumin, another synthetic curcumin derivative, significantly inhibited expression of MMP-2 and -9 by inhibition of the STAT3 signaling pathway as presented in breast cancer cell lines [55]. A very promising modification of curcumin aims at improving the binding of zinc ions. A novel MMP inhibitory formulation combines tetracyclines with curcumin [50]. Tetracycline molecules contain diketonic moiety, which binds zinc ions, and thus, may inhibit zinc-containing MMPs irrelevantly to antibacterial properties [54]. Similar metal-binding moieties contain curcumin molecules, and thus, complexing tetracycline with curcumin will combine and increase their inhibitory effect on MMPs by the zinc-binding properties of both being formulated into a new compound [54,56].

## 6. The Rationale for Curcumin Application in Bladder Cancer as a Potential Factor Limiting the Progression of the Disease

The characteristic clinical feature of bladder cancer is its propensity to recur and progress. Approximately three-quarters of cases are initially confined to the urothelium, not crossing the BM, as non-muscle-invasive cancer (NMIBC). The remaining one-quarter of cases manifest as muscle-invasive neoplasm (MIBC). A five-years recurrence rate of NMIBC reaches 70%, and one-third of them will progress to invasive MIBC. At the stage of invasive disease, the prognosis is much worse, as 15% patients will develop metastases [50,83,84]. The response rate for combined chemotherapy such as MVAC (methotrexate + vinblastine + doxorubicin + cisplatin) reaches up to 20%; however, 60% of patients do not respond to the treatment [51]. 

Surgical treatment of NMIBC is complemented by intravesical immunotherapy or chemotherapy with attenuated tuberculosis mycobacteria (BCG) or cytostatics (mitomycin), respectively. The aim of this treatment is to prevent the disease from becoming the muscle-invasive form. Immunotherapy and chemotherapy show efficacy, but a significant percentage of patients do not respond to such treatment, and develop severe adverse reactions. Therefore, research is targeted at the development of innovative high-potential medications replacing or enhancing the above adjuvant therapies to significantly improve the effectiveness of treatment [50,85]. Unfortunately, existing and innovative systemic and intravesical drugs are characterized by a high rate of side effects, along with a failure to achieve expected outcomes. Over 30% of patients do not respond to intravesical BG treatment [85]. Intravesical chemotherapy (mitomycin, epirubicin, thiotepa) causes local and systemic toxicity, whereas its actual therapeutic efficacy is limited [84]. Up to 50% of patients respond to adjuvant chemotherapy, but almost all are affected by its toxicity [83,85]. 

Natural plant-derived products have emerged as valuable alternatives with proven nontoxicity and suitability for the prevention and treatment of a number of ailments. Yet, only recently, meticulous research has revealed the complexity of the molecular mechanisms of action of commonly used nutraceuticals. Curcumin is an extensively tested nutrient for its proven, virtually unlimited therapeutic qualities. Its beneficial effects have been shown for a number of human malignancies during several clinical trials. A small series of tests have been carried out on bladder cancer. As bladder cancer therapy is the most capital-intensive among all cancer therapies, it is reasonable to reach for inexpensive, but effective, substances complementing and amending surgery—still accepted as the contemporary essential form of the treatment [86,87]. Existing clinical and laboratory experiments present multidirectional anticarcinogenic activities of curcumin on bladder cancer as monotherapy, or synergistically potentiating other chemotherapeutics. Moreover, experimental data presents curcumin bioactivity on all stages of carcinogenesis both intra- and extracellularly at concentrations of at least 40 μmol/L [86]. A number of experiments were conducted in vitro on experimental cell lines to explain intracellular mechanisms of curcumin activity [85]. Yet, despite having obtained many data, the exact mechanisms by which curcumin executes observed effects have not been fully explained [86,87]. 

Clonal assays presented and proved that curcumin is lethal to BC cell lines inducing apoptosis and arrest of the cell cycle in both G1/S and G2/M phases [85,88]. Curcumin also downregulated expression of essential antiapoptotic proteins (Bcl-2, Survivin, NF-κB) in parallel with upregulation of proapoptotic mediators (Bax, p53, caspase 3). These effects are stronger than those caused by cisplatin alone [88,89,90]. The apoptotic effect of gemcitabine and paclitaxel was intensified when co-administered with curcumin [81,85,89]. Concurrently, curcumin exhibits chemopreventive properties, due to in-vitro-presented inhibition of intracellular pathways activating external chemical carcinogens [84]. 

Still, intravesical BCG immunotherapy remains the most effective method of treatment of bladder cancer. Prevention of tumor recurrence and of transition from non-muscle invasive to invasive tumor is the main goal of such therapy. Several in vitro and in vivo studies presented prominent synergistic effects of co-administration of BCG and curcumin [77,78]. This beneficial phenomenon is multidirectional through influence on different cellular signaling molecules. Curcumin potentiates proapoptotic effects of the tumor necrosis factor-related apoptosis-inducing ligand (TRAIL) by upregulation of its main DR5 membrane receptor [91,92]. Because BCG acts mainly through stimulation of the expression of TRIAL by neutrophils, co-administration with curcumin will enhance and intensify the BCG-induced immune response in bladder cancer [77,78,91]. On experimental cancer cells lines, the percentage of apoptotic cells ranged from 43% to 74% for BCG alone and BCG+curcumin, respectively [91]. The ineffectiveness of BCG-therapy observed in a subset of patients can be explained by the TRAIL resistance of tumor cells, related to constitutive (over)activation of nuclear factor NF-κB [91]. As curcumin suppresses NF-κB and further expression of dependent genes, its addition to BCG can reverse this resistance, and potentiate apoptosis [92]. 

Another pathway contributing to BCG-therapy failure is the overexpression of proapoptotic protein Bcl-2, which inhibits TRAIL-induced apoptosis. As described, curcumin effectively inhibits Bcl-2 and enhances tumor cells death [93]. In vitro assays proved that expression of proliferation marker cyclin D1 decreased by 50% after treatment with BCG alone, and up to 90% concomitantly with curcumin [91]. A proapoptotic response was additionally supported, and intensified by the formation of reactive oxygen species under curcumin influence [85,86]. Several other cellular pathways are modulated by curcumin, such as an inhibition of nitric oxide synthase, tyrosine kinases, transcriptional factors c- jun/AP-1, arachidonic acid pathways, COX activity, and many more. Wnt/Beta-catenin signal transduction pathways are also targeted by curcumin by downregulation of catenin, thus, affecting crucial metastasis EMT induction in bladder cancer cells [86]. This activity overlaps with the inhibition of MMP-2 and -9 expression, also involved in EMT processes. Indeed, the expression of mesenchymal markers (vimentin, N-cadherin) decreases in the presence of curcumin in a dose-dependent manner, whereas the expression of epithelial differentiation markers (E-cadherin) increases. This indicates that curcumin suspends bladder cancer cells in an epithelial, polarized phenotype; and restricts ″mobile″ mesenchymal features, hampering cancer cell migration, bladder cancer invasiveness, and further metastasis [94]. Curcumin also suppresses beta-catenin overexpression in bladder cancer cells, and thus, reverses metastatic potential and migration of bladder cancer cells in a dose-dependent manner [86,87].

The above data and other published studies suggest that the suppression of the NF-κB pathway is most likely essential for the multidirectional activity of curcumin (anti-inflammatory, anti-proliferatory, anti-invasive, anti-angiogenic) [89,91,92]. This mechanism was depicted in a previous part of this paper, and was proved on experimental bladder cancer cell lines [89]. The evidence shows that several others pathways associated with PI3K/AKT/mTOR, ERK1/2-signaling, insulin receptor substrate-1, insulin-like growth factor-2, and trophoblast cell surface antigen-2 related to bladder carcinogenesis are also affected by curcumin [50,51]. Data from in vitro experiments are confirmed by in vivo studies. In animal models of human bladder cancer, both xenografts and chemically induced curcumin inhibited cancer cell implantation, tumor growth, and metastasis [86]. These effects were observed after gavage, and after intravesical instillation. To overcome poor intestinal absorption, alternative modes of application have been tested. In line with intravesical administration, intraperitoneal and intravenous injections proved to be systematically effective and safe in a rat bladder cancer model [84]. All above studies show that curcumin is an effective modality in the prevention of NMIBC bladder cancer recurrence and progression, both as a sole agent and as a synergistic additive chemosensitizer for existing therapies [50,51]. 

Intravesical instillations of curcumin in experimental mouse bladder cancer models resulted in tumor necrosis and a significant reduction of tumor size, but with no effect on the number of tumors per bladder [89]. The combination of curcumin and BCG appeared to be more effective than BCG or curcumin alone [91]. In addition to the impact on existing tumors, curcumin inhibited implantation of free cancer cells in a bladder cancer murine model through the influence on integrin adhesion receptors [95,96]. A beneficial additive effect has been presented for intravesical or intratumoral or oral administration of curcumin concurrent with standard intravesical administration of BCG, with doses up to 8000 mg/day [91]. Even though only a few such studies have been published, evidences suggest a high therapeutic capacity of curcumin in bladder cancer, and place curcumin as possible integral part of therapy [50,51]. The synergy of BCG and curcumin is reflected in cellular molecular events, such as in a decrease of cell proliferation proteins (Ki67, cyclin D1, c-myc), anti-apoptotic proteins (Bcl-2, Bcl-xl, survivin), and pro-angiogenic proteins (CD31, VEGF), and the inhibition of epigenetic controllers (HDAC) [51]. 

Curcumin also potentiates therapeutic effects, and alleviates side effects of bladder cancer adjuvant chemotherapy. It increases therapeutic efficiency, and reverses tumor resistance to gemcitabine [97]. The combination of oral curcumin with intraperitoneal cisplatin resulted in an increased therapeutic efficiency of chemotherapy, and an increased reduction of tumor size in mouse models [98]. This synergistic effect can be explained by the induction of reactive oxygen species-related activation of proapoptotic pathways, and the inhibition of antiapoptotic pathways [98]. Concomitantly, curcumin alleviates cisplatin-related nephrotoxicity.

Acute kidney injury is the most common severe side effect of cisplatin-based therapy. Nephrotoxicity is related to a series of dysfunctional cellular metabolic pathways which pathologically intensify renal inflammatory processes, along with devastating oxidative stress, necrosis, and apoptosis of proximal tubular epithelial cells (PTEC) [51,99,100,101]. Cisplatin directly and/or indirectly induces upregulation of key molecules involved in those pathways, among others: TNF-α, p53, Fas ligand/receptor system, COX-2, caspases, and nuclear transcription factor-kappa B (NF-κB). These pathways are interlinked through common molecules [99,100,101]. Overexpression of TNF-α activates humoral and cellular inflammatory processes, and induces the generation of reactive oxygen species, leading to renal damage. P53 activates Bax-related proapoptotic pathways, and impairs Bcl-2- and Bcl-xL-related antiapoptotic pathways. Also, the Fas ligand/receptor apoptotic system is pathologically induced by cisplatin-upregulated p53 [99,100,101]. Cisplatin-related activation of transcription factor NF-κB, together with upregulated p53, stimulates pathways suppressing nephroprotective cytokine HNF1β, and also through the promotion of TNF-α synthesis [99,101]. Such interactions induce renal tubular cells apoptosis and necrosis, resulting in acute kidney injury. Further, epithelial damage attracts an influx of immunocompetent cells, with subsequent aggressive renal damage. Studies revealed that selective suppression of the above molecules alleviates cisplatin-related tubular epithelium dysfunction and renal injury. Existing nephroprotective strategies (cimetidine, mannitol, amifostine, celecoxib, etc.) are not always effective [51,99,100]. Curcumin directly and indirectly blocks the aforementioned molecules and involved pathways (see Figure 2). Laboratory experiments clearly present that curcumin reduced acute kidney injury in mice, and downregulated the pro-apoptotic cisplatin-related response in renal tubular cells [99,101]. This curcumin-associated reno-protective effect is achieved precisely by targeting key multifunctional cytokines, such as p53, NF-κB, Bax, and the others mentioned above. Therefore, curcumin appears to be an important renoprotective complementary and supportive agent for cancer chemotherapy [99].

Immunologic escape of the tumor has also been modulated by curcumin through inhibition of expression of programmed cell death ligand 1 (PDL1) on both bladder cancer cells and tumor-infiltrating lymphocytes, as proved by in vivo and in vitro experiments [51]. Also, a clinical trial presented that curcumin intensified patients’ immunologic response by stimulation of interferon-gamma production, and the propagation of T- helper lymphocytes and cytotoxic NK cells [51]. 

To fully exploit anti-PDL1 activity of curcumin, a few studies have been conducted. The combined application of curcumin molecules and anti-PDL1 antibodies was tested in a bladder tumor mouse model [51]. This construct targets the programmed cell death protein (PD-1) receptor of lymphocytes. One clinical trial on patients with gynecological carcinomas combining pembrolizumab (clinically approved anti-PD1 receptor antibody) with RTG therapy and curcumin food supplementation (with additional vit. D, lansoprazole, aspirin, cyclophosphamide) was conducted to elevate the proportion of response for a PD-1-blockade + radiotherapy treatment regime [102].

## 7. Curcumin as an Inhibitor of MMP-2 and MMP-9, Restricting Progression of Bladder Cancer: Rationale, Perspectives, and Obstacles to Overcome

Constraining non-muscle-invasive bladder cancer transition into the muscle-invasive form seems to be crucial for successful treatment. Bladder cancer invasiveness can be compromised by interference into MMP activity and processes remodeling the urothelium basement membrane and underlying connective tissue. The role of MMP-2 and -9, and their tissue inhibitor, TIMP-2, have been investigated in several human neoplasms for their supposed role in cancer invasiveness. Their expression in bladder cancer tissues is described in a previous part of this paper. Moreover, despite sometimes unclear scientific research data on the relationship between the expression of MMP and the stage of disease, MMPs can be used as a predictive/prognostic factor in bladder cancer. As curcumin shows inhibitory activity on MMP expression in several neoplasms both in vitro and in vivo, it is rational to harness this phenomenon in bladder cancer. The results from research using curcuminoids in vitro in different neoplasmatic cell lines, in vivo on animal cancer models, and from limited human trials are definitely encouraging [103]. Those conclusions and data can be extrapolated and used on bladder cancer patients. 

Yet, studies in relation to MMPs in bladder cancer and curcumin are scarce and still under investigation [104]. An in vitro study on human bladder cancer cell lines (T24, 5637) treated with curcumin indicated an inhibition of cell migration through interference with MMPs [101]. In both cell lines, the expression of MMP-2 and MMP-9 degrading the basement membrane significantly decreased, whereas the expression of inhibitors of metalloproteinases significantly increased [101]. This indicates a reduced metastatic potential of bladder cancer cells in an environment supplemented with curcumin. Also, the suppression of metalloproteinase-related signaling pathways resulted in an increase of apoptosis of analyzed cell lines. As MMPs are involved in invasion and metastasis, it can be assumed that in an environment enriched with curcumin, the risk of transition from non-muscle-invasive to muscle-invasive tumor should be significantly diminished [51]. This conclusion is supported by the observation that intravesically-administered curcumin prevented implantation of bladder cancer cells in a mouse tumor model [95]. 

Still, some obstacles exist. To fully exploit curcumin anticancer properties for the treatment of bladder cancer, an appropriate concentration must be maintained in tumor tissue and the surrounding microenvironment. This is a critical curtailment. Unfortunately, due to above-described biochemical properties of curcumin, even large doses applied orally up to 8 g/day did not result in detectable concentrations in urine [50]. Even extremely high, long-administered doses resulted in a modest serum concentration of curcumin, considerably lower than concentrations at which anticancer effects were observed in cell lines or animal models (>10 microM) [105]. Peak serum concentrations in patients after oral ingestion of 8 g/day ranged up to 2 microM, without detectable amounts in urine [51,85]. However, it should be stressed that such high doses did not cause significant side effects other than diarrhea or abdominal fullness (14% and 29% of patients, respectively) [51]. Rats fed with curcumin in doses up to 5 g/kg also did not show toxic side effects [85]. In vitro experiments with few different bladder cancer cell lines demonstrated a pro-apoptotic effect of curcumin only at a concentration 40 microM [85]. Paradoxically, low concentrations (1–10 microM) may have an anti-oxidative protecting effect on tissues, and only higher concentrations have a pro-oxidative effect and promote cell death [51]. Also, an anti-angiogenic effect was observed at lower concentrations, ranging from 10–30 microM [106]. 

Other than the proven complementary action, the additional effect of combined use of curcumin plus chemotherapeutics is its chemopreventive activity [84]. Several studies with different combinations of current compounds show a significantly reduced toxicity of a number of chemotherapy regimens with fluorouracyl, gemcidabine, paclitaxel, doxorubicin, BCG, etc. [51,76,78]. 

In relation to the worldwide ongoing pandemic situation, it should be added that a number of works have shown the effectiveness of turmeric in the treatment of severe viral infections, including the most severe forms of COVID-19 [107,108,109,110].

## 8. Remarks on Curcumin and Other Plant-Derived Drugs in the Light of Clinical Trials

Naturally occurring plant-derived medicines are back in the spotlight of modern medicine. Raw plant extracts have been used in traditional medicine, but, recently, several pure compounds or their chemical modifications have been extensively tested for the treatment of benign and malignant diseases. Their wide range of applications has been thoroughly explored in the scientific literature [111].

A plethora of in vitro (on cell lines) and in vivo (on animal models) research has provided evidence that phytochemicals reveal multidirectional anti-cancer effects on cancer cells [51,111,112]. Clinical and preclinical studies on the use of different phytodrugs in the treatment of human malignancies and benign chronic diseases have been published [111,113,114,115].

Among these medications, curcumin is the most extensively clinically tested [51,114]. Randomized controlled trials proved therapeutic potential of curcumin against a number of chronic benign human diseases (Alzheimer’s disease, acute coronary syndrome, bowel diseases, diabetes, psoriasis, cardiovascular diseases, depression, metabolic syndrome, hypertriglyceridemia, osteoarthritis, fatty liver disease, etc.) [73,115]. Similarly, promising results have been obtained against a number of cancers (colorectal, pancreatic, prostate, lung, oral, head and neck cancers, and osteogenic sarcoma) [73,112,115]. Doses of curcumin widely ranged from 20 mg/day to 3 g/day, and even extremely high doses up to 15 g/day [112,115]. Yet, clinical data on the application of curcumin in bladder cancer are limited. The official webpage, clinicaltrials.gov, reports 75 clinical trials on curcuminoids for a variety of cancers, but none related to its use in bladder cancer (data refers to December 2021) (see Table 1). Apparently, ongoing or completed clinical investigations are so far missing. However, some preliminary research has been published, although at an early stage and with a limited number of participants, and data are all preliminary or as a preclinical study in combination with intravesical BCG-therapy [73,78].

## 9. Conclusions

All the research conducted presents curcumin as a valuable, potent, natural phytochemical compound with outstanding therapeutic properties for human cancers. A number of in vitro, animal in vivo, and human clinical trials proved its multidirectional effectiveness in the treatment of bladder cancer as a cancer-preventive, therapeutic, protective, and chemosensitizing modality [52,53,54,55]. The novelty of this treatment combines perfectly with its easy availability, lack of toxicity, and minimal financial burden. In addition to impacting multiple metabolic and signaling cellular pathways of bladder cancer, curcumin stabilizes intratumoral milieu and the surrounding microenvironment through an inhibitory effect of matrix metalloproteinases degrading bladder wall connective tissue [34,57,58,59]. All crucial hallmarks of bladder cancer development are affected by curcuminoids. This has been proven in the works cited above [57,58,59,60,61,62,63,64,65,66,67,68,69,70,71,72,73,74,75,76,77,78,79,80,81,82,83,84,85,86,87,88,89,90,91,92,93,94,95,96,97,98,99]. Preservation of the integrity of the urinary bladder wall connective tissue elements is a key prerequisite for the containment of cancer progression, the restriction of deep invasion, and the prevention of transition from the non-muscle-invasive to the deadly muscle-invasive form. Undeniably, natural products, especially curcumin and its derivatives, deserve to become a valuable element of modern clinical practice. Large-scale human trials on the usefulness of curcumin are indispensable for gathering clinical data supporting the promising results of experimental studies. 

## Figures and Tables

**Figure 1 nutrients-14-00032-f001:**
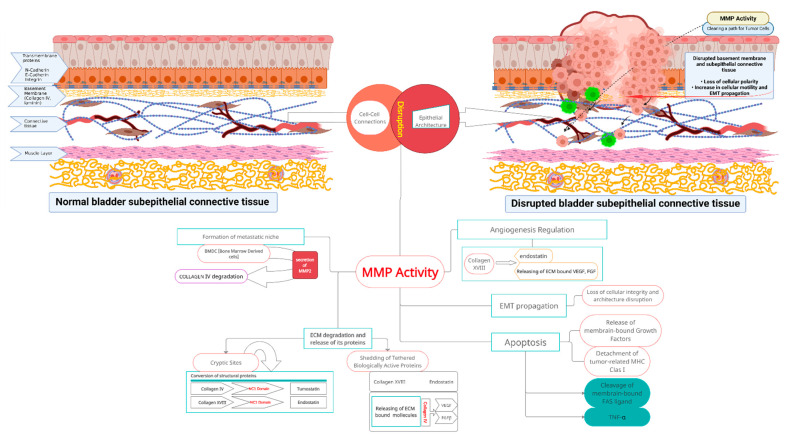
MMP activity and related molecular events during bladder cancer propagation. **Upper left part of the diagram:** Normal bladder wall structure. Correct epithelial architecture and polarity provided by the indicated adhesion molecules, and with a normal basement membrane. Layered self-assembled collagen IV and laminin networks blend intact basement membrane (yellow). Regular structure of underlying connective and muscle tissues. Intact layered structure of bladder wall is crucial in proper bladder physiology. **Upper right part of the diagram:** Gradual bladder wall disruption related to aberrant MMP activity. Cleavage of intercellular junctions, along with basement membrane remodeling and further loss of polarity of the urothelium, increased cell motility with propagation of epithelial-to-mesenchymal transition. Cancer cells traverse degraded supporting barrier (formerly continuous basement membrane). **Center of the diagram:** Degradation of extracellular matrix mediated by matrix metalloproteinases 2 and 9 released by cancer cells, and accompanying cells from tumor-associated microenvironment cleave components of the basement membrane and underlying connective tissue, and generate new derivative molecules. As a result, extracellular matrix structural proteins and matrix-bound latent signaling molecules are converted into biologically active signaling molecules that are converted into active mediators. Activity of those products (products listed in the following segments of central part of the scheme) results in further cancer cells chemoattraction, proliferation, infiltration of deep tissues, aberrant neo-angiogenesis, inhibition of apoptosis in cancer cells, and formation of cancer niches. These molecular events contribute to bladder cancer progression from non-muscle to muscle-invasive tumors. MMP actions that take place after MMP activation are explained in the body text.

**Figure 2 nutrients-14-00032-f002:**
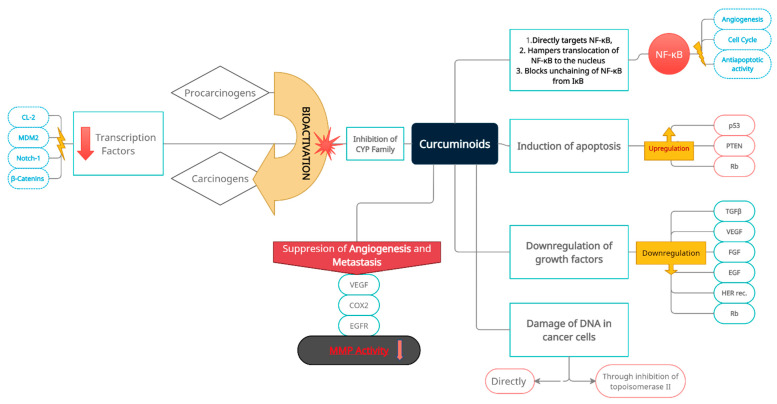
Multidirectional and multilevel preventive and inhibitory effects of curcuminoids on oncogenesis, with a special emphasis on induction and progression of bladder cancer, and its subsequent development. Prevention and inhibition of neoplastic transformation and progression is achieved through induction of apoptosis, inactivation of cancer-related transcriptional pathways (nuclear transcription factors and related oncogenes), cancer cell DNA damage (direct and indirect), inactivation of extracellular carcinogens, downregulation of growth factors, activation of cancer-suppressing genes, suppression of angiogenesis, and metastatic niche formation. Consecutive boxes of the diagram group intra- and extracellular molecular targets of curcumin.

**Table 1 nutrients-14-00032-t001:** A summary of active and completed clinical trials studying curcumin in cancer treatment and prevention.

	Official Title of Study	Disease	Number of Patients	Institution	Phase	Study Design	Administration and Dosage of Curcumin	Start	ClinicalTrials.gov Identifier:
1	Avastin/FOLFIRI in Combination with Curcumin in Colorectal Cancer Patients with Unresectable Metastasis	Colorectal Cancer	50	Gachon University Gil Medical Center	Phase 2	Bevacizumab/FOLFIRI in Combination with Curcumin	Oral, 100 mg	August, 2015	NCT02439385
2	Curcumin Chemoprevention of Colorectal Neoplasia	Colorectal Cancer	40	University of North Carolina, Chapel Hill	Phase 1	Curcumin	Oral, 4000 mg	November 2010	NCT01333917
3	A Pilot, Feasibility Study of Curcumin in Combination with 5FU for Patients with 5FU-Resistant Metastatic Colon Cancer	Colorectal Cancer	13	Baylor Charles A. Sammons Cancer Center	Phase 1	CurcuminIn Combination with 5-Flurorouracil	Oral, 500 mg	March 2016	NCT02724202
4	A Prospective Evaluation of the Effect of Curcumin on Dose-limiting Toxicity and Pharmacokinetics of Irinotecan in Colorectal Cancer Patients	Colorectal Cancer	23	University of North Carolina at Chapel Hill Lineberger Comprehensive Cancer Center	Phase 1	Curcumin + Irinotecan	Oral, 4000 mg,	June 2013	NCT01859858
5	A Randomized Double Blinded Study of Curcumin with Pre-operative Capecitabine and Radiation Therapy Followed by Surgery for Rectal Cancer	Colorectal Cancer	45	M.D. Anderson Cancer Center	Phase 2	Capecitabine + Curcumin vs. Capecitabine + placebo	-	August 2008	NCT00745134
6	A Phase I/IIa Study Combining Curcumin (Curcumin C3-Complex, Sabinsa) with Standard Care FOLFOX Chemotherapy in Patients with Inoperable Colorectal Cancer.	Colorectal Cancer	41	Dept Oncology, Leicester Royal Infirmary, University Hospitals Leicester	Phase 2	Curcumin + Chemotherapy—FOLFOX	Oral, 2000 mg	February 2012	NCT01490996
7	Meriva for Treatment-Induced Inflammation and Fatigue in Women with Breast Cancer	Breast Cancer	30	Emory Winship Cancer Institute	Phase 2	Curcumin vs. Placebo	Oral, 100 mg	May 2015	NCT01740323
8	Effect of Preoperative Curcumin in Breast Cancer Patients	Breast Cancer	30	University of Malaya	N/A	Curcumin vs. Placebo	Oral, 8000 mg	June 2017	NCT03847623
9	Study of Efficacy of Curcumin in Combination with Chemotherapy in Patients with Advanced Breast Cancer: Randomized, Double Blind, Placebo Controlled Clinical Trial	Breast Cancer	150	National Center of Oncology, Armenia	Phase 2	Curcumin + Paclitaxel vs. Paclitaxel + Placebo	Parenteral, 300 mg	March, 2017	NCT03072992
10	Nanoemulsion Curcumin for Obesity, Inflammation, and Breast Cancer Prevention—A Pilot Trial	Breast Cancer	29	Ohio State University Comprehensive Cancer Center	N/A	Curcumin	Oral, 100 mg	June 2013	NCT01975363
11	Radiation Therapy with or without Curcumin Supplement in Treating Patients with Prostate Cancer	Prostate Cancer	40	Oncology and Radiotherapy Department, Besat Hospital, Tehran	N/A	Curcumin vs. Placebo	Oral, 3000 mg	March 2011	NCT01917890
12	Phase II Trial of Curcumin in Patients with Advanced Pancreatic Cancer	Prostate Cancer	50	M.D. Anderson Cancer Center	Phase 2	Curcumin	Oral, 8000 mg	November 2004	NCT00094445
13	Phase II Study of Nanocurcumin Versus Placebo for Patients Undergoing Radiotherapy for Prostate Cancer	Prostate Cancer	64	Shahid Beheshti University of Medical Sciences	Phase 2	Curcumin vs. Placebo	Oral, 120 mg	March 2016	NCT02724618
14	Phase II Trial of Gemcitabine and Curcumin in Patients with Advanced Pancreatic Cancer	Pancreatic Cancer	17	Rambam Health Care Campus	Phase 2	Curcumin + Gemcitabine	Oral, 8000 mg	July 2004	NCT00192842
15	An Exploratory Biomarker Trial of the Food Substances Curcumin C3 Complex in Subjects with Newly Diagnosed Head and Neck Squamous Cell Carcinoma	Head and Neck Carcinoma	33	Feist-Weiller Cancer Center at Louisiana State University Health SciencesNational Cancer Institute (NCI)	Phase 1	Curcumin	Oral, 8000 mg	June 2010	NCT01160302
16	Effect of Curcumin Addition to Standard Treatment on Tumor-Induced Inflammation in Endometrial Carcinoma	Endometrial Carcinoma	7	University Hospital KU Leuven Campus Gasthuisberg	Phase 2	Curcumin	Oral, 2000 mg	October 2013	NCT02017353
17	Randomized, Double-Blind, Placebo-Controlled Trial of Meriva^®^ (Curcuminoids) as a Candidate Chemoprevention Agent for Gastric Carcinogenesis	Chronic Atrophic Gastritis/Gastric Cancer	100	National Cancer Institute (NCI), Mayo Clinic in Rochester	Phase 2	Curcumin vs. Placebo	Oral, 200 mg	April 2017	NCT02782949
18	Pilot Study of Curcumin (Diferuloylmethane Derivative) with or without Bioperine in Patients with Multiple Myeloma	Multiple Myeloma	42	M.D. Anderson Cancer Center	N/A	Curcumin vs. Curcumin + Bioperine	Oral, 2000 mg	November 2004	NCT00113841

## Data Availability

The data presented in Table 1 are openly available in ClinicalTrials.gov (accessed on 17 December 2021).

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
