# Peer review of "Curcumin May Prevent Basement Membrane Disassembly by Matrix Metalloproteinases and Progression of the Bladder Cancer"

_nutrients, 2021, doi:10.3390/nu14010032_

Round 1
Reviewer 1 Report
This review focises on a topic of particular relevance on the use of curcumin in the treatment of bladder cancer.
I think however that the review can and should be shortened of at least 1/3 and even more in the introductory part before discussing curcumin. In addition the paper needs a singificant linguistic revision.
I have also seen that no clinical studies on the use of curcumin in bladder cancery has been registered in clinical trial.gov while there are studies of curcumin in other forms of cancer. If this is so, the Authors shoul indicate that only preclinical studies are conducted on the potential beneficial effects on the role of curcumin in bladder cancer and ongoing or completed clinical investigations are so far missing.
I would recommend that the Authors quote this recent review on the topic of natural compounds in cancer
Biotechnol Adv. 2018 Nov 1;36(6):1622-1632.
Along this line, the beneficial effects of curcumin in several in vitro and in vivo models of cancer should be discussed . i would also propose the Authors to prepare a table, if feasible, on the ongoing or completed clinical studies of curcumin in cancer (from clinical trial gov)
Author Response
Dear Editors and Reviewers,
thank You for Your work and Your time. I am grateful to the reviewers for their critical comments and useful suggestions that have helped me to improve submitted paper. As indicated in the responses that follow, I have taken all these comments and suggestions into account in the revised version of my paper.
Reviewer 1
Thank You for Your kind reading of the submitted manuscript.
Thank You very much for Your suggestions and comments regarding this manuscript.
Changed, added and supplemented texts are highlighted in red.
- The review can and should be shortened of at least 1/3 and even more in the introductory part before discussing curcumin. In addition the paper needs significant linguistic revision.
Response:
Following the reviewer's rightful suggestion, the introduction has been shortened from 637 words to 339 words. An abstract has been completely rewritten. Some parts of the text have been merged together to avoid repetition of similar issues. Dear Reviewer, I have taken the liberty of not shortening the section on the role of basement membrane and enzymes because a detailed description of these issues is necessary to understand the stages of bladder cancer development and the role played by metalloproteinases. I kindly ask You to consider this assumption and to adopt my (suggested) point of view. I understand that as a non-native speaker I have committed a linguistic irregularity. Language proofreading was done according to the instructions of a professor of medicine at the University of Toronto. The title and sections of the article have been redrafted for linguistic correctness and clarity.
- No clinical studies on the use of curcumin in the bladder cancer has been registered in ClinicalTrial.gov while there are studies of curcumin in other forms of cancer. Authors should indicate that only preclinical studies are conducted on the potential beneficial efects on the role of curcumin in the bladder cancer and ongoing or completed clinical investigations are so far missing.
Response:
I completely agree with Reviewer 1. I have corrected this deficiency by adding an additional paragraph under the title „Remarks on curcumin and other plant-derived drugs in the light of clinical trials” to properly explain the suggested issue. Links to relevant scientific literature have been added.
- I would recommend that the Authors quote this recent review on the topic of natural compounds in cancer Mijatović S, Bramanti A, Nicoletti F, Fagone P, Kaluderović GN, Maksimović-Ivanić D. Naturally occurring compounds in differentiation based therapy of cancer. Biotechnol Adv. 2018
Response:
Thank you for bringing this scientific material to my attention. The reviewer's rightly suggested scholarly publication was included in the added paragraph „Remarks on curcumin and other plant-derived drugs in the light of clinical trials”
- The beneficial effects of curcumin in several „in vitro” and „in vivo” models of cancer should be discussed. I would also propose the Authors to prepare a table on the ongoing or completed clinical studies of curcumin in cancer (from ClinicalTrial.gov)
Response:
This comment contributed to the completion of the text and presentation of curcumin's therapeutic potential. The favorable results obtained during these studies against a number of cancers are further highlighted in the section titled” Biological properties of curcumin in cancer processes”. Following the reviewer's suggestion, a table showing the relevant clinical trials has been prepared.The table has been sent as a separate file to be included at the end of the paper.
Reviewer 2 Report
- A revision of the grammar will be useful. At some points disappear some subjects, some articles are missed and some of them are unnecessary.
- Words like “in vitro” o “in vivo” or “C. longa” require quotation marks or italics. Please check it.
- Section 5 line 281. The authors point that curcumin shows no toxicity. Can the authors provide any reference regarding this statement?
- Lines 365-368 address the problems related to the “in vivo” effect of curcumin. Nevertheless, one important point is missed: the low solubility. This low solubility is the main reason for the los bioavailability. Please add some information regarding this problem
- Lines 369-379 illustrate different strategies to deliver curcumin with “in vivo” effect. However, cyclodextrins are not mentioned in this paragraph. Also, some information about the different formulation commercialized may be useful (Meriva, Curarti…)
- Section 6, lines 432-448. The benefits of curcumin therapy are shown in different studies. Nevertheless, the dose use is missed. Please include it.
- Line 530. Please can the authors provide a possible mechanism involved in cisplatin-related nephrotoxicicy alleviation?
- Lines 593-605. This information sees to repeat some statement from lines 370-379. Please consider to “merge” this information.
Author Response
Dear Editors and Reviewers,
thank You for Your work and Your time. I am grateful to the reviewers for their critical comments and useful suggestions that have helped me to improve submitted paper. As indicated in the responses that follow, I have taken all these comments and suggestions into account in the revised version of my paper.
Reviewer 2
Thank You for Your kind reading of the submitted manuscript.
Thank You very much for Your suggestions and comments regarding this manuscript.
Changed, added and supplemented texts are highlighted in red.
- A revision of the grammar will be useful. At some points disappear some subjects, some articles are missing and some of them are unncecessary
Response:
Thank you for pointing out these problems. Some issues - indicated further by the honourable reviewer - have been supplemented. Some of the source literature has been removed, while new items have been added in order to better highlight the important and added points of the text. I understand that as a non-native speaker I have committed a linguistic irregularity. Language proofreading was done according to the instructions of a professor of medicine at the University of Toronto. The title and sections of the article have been redrafted for linguistic correctness and clarity.
- 2. Words like „in vitro” or „in vivo” or. C.longa” require quotation marks or italics.
Response:
Thank You. This oversight has been corrected.
- Section 5 line 281. The authors point that curcumin shows no toxicity. Can authors provide any refernce regarding this statement?
Response:
Indeed, this was missing from the text.The following literature items have been added for completeness
- Rutz, J.; Janicova, A.; Woidacki, K.; Chun, F.K.; Blaheta, R.A.; Relja, B. Curcumin-A viable agent for better bladder cancer t Int J Mol Sci, 2020, 21, 3761. doi: 10.3390/ijms21113761.
- Chainani-Wu, Safety and anti-inflammatory activity of curcumin: a component of tumeric (Curcuma longa). J Altern Complement Med, 2003, 9, 161-168. doi: 10.1089/107555303321223035.
- Kocaadam, B.; Sanlier, Curcumin, an active component of turmeric (Curcuma longa), and its effects on health. Crit Rev Food Sci Nutr, 2017, 57, 2889-2895. doi: 10.1080/10408398.2015.1077195.
- Lines 365-368 address the problems related to the „in vivo” effect of curcumin. Nevertheless, one importand point is missed: the low solubility. This low solubility is the main reason for the low bioavailability. Please ad some information regarding this problem
Response:
Information regarding the issue of low bioavailability related to low water solubility has been added in the space indicated by the reviewer in section: „Biological properties of curcumin in cancer processes”. This explanation has been combined into one new supplemental segment with explanations and additions relevant to the reviewer's comments located in items 4, 5, and 8. Relevant literature has also been added.
- Lines 369-379 illustrate different strategies to deliver curcumin with „in vivo” effect. However, cyclodextrins are not mentioned in this paragraph. Also, some information about the different formulation commercialized may be useful (Meriva, Curanti).
Response:
Thank you for pointing this out. Indeed, this issue needed to be supplemented with the formulations indicated by the reviewer. This explanation has been combined into one new supplemental segment with explanations and additions relevant to the reviewer's comments located in items 4, and 8. Relevant literature has also been added.
- Section 6, lines 432-448. The benefits of curcumin therapy are shown in different studies. Nevertheless, the dose use is missed. Please include it.
Response:
This has been added where indicated by the Reviewer along with a reference to the relevant literature. However, I dare note that the dosages and concentrations used by the researchers are also given in various places in the article. Nevertheless, the reviewer's rightful remark will facilitate understanding of the given fragment of the article.
- Line 530. Please can the authors provide a possible mechanism involved in cisplatin-related nephrotoxicity alleviation?
Response:
The explanation was placed from line indicated by the reviewer. This issue required additional expansion of the text to include a brief description of the mechanism of cisplatin nephrotoxicity. I dared to include such information in a shortened version. I have also described what - in the light of the mechanism described above - may be the mechanism of protection of kidneys by turmeric. Links to relevant literature have also been added.
- Lines 593-605. This information sees to repeat some statement from lines 370-379. Please conside to „merge” this information.
Response:
Thank you for pointing this out. In fact, the indicated portions of the text contained repetition and have been consolidated into a new added segment along with the issues related to comments 4 and 5. Relevant literature has also been added. I hope that such presentation of remarks mentioned in points 4,5 and 8 will meet reviewer's expectations
Round 2
Reviewer 1 Report
The Authors have adequately addressed my criticisms
Author Response
Thank you for your help and contribution!
Reviewer 2 Report
Well done!
Author Response

(The authors gave the same response as above.)
